# Anti-Fouling Properties of Phosphonium Ionic Liquid Coatings in the Marine Environment

**DOI:** 10.3390/polym15183677

**Published:** 2023-09-06

**Authors:** Sajith Kaniyadan Baiju, Brent James Martin, Rayleen Fredericks, Harikrishnan Raghavan, Karnika De Silva, Matthew Greig Cowan

**Affiliations:** 1Department of Chemical and Process Engineering, University of Canterbury, Private Bag 4800, Christchurch 8140, New Zealand; sajith.kaniyadanbaiju@pg.canterbury.ac.nz (S.K.B.); rayleen.fredericks@canterbury.ac.nz (R.F.);; 2New Zealand Product Accelerator, University of Canterbury, Private Bag 4800, Christchurch 8140, New Zealand; 3MacDiarmid Institute for Advanced Materials and Nanotechnology, University of Canterbury, Private Bag 4800, Christchurch 8140, New Zealand; 4Defence Technology Agency (DTA), Private Bag 32901, Auckland 0744, New Zealand; 5NZ Product Accelerator, Faculty of Engineering, University of Auckland, Auckland 1010, New Zealand

**Keywords:** anti-fouling, marine anti-fouling, biofouling, ionic liquids, hydrophobicity, trialkylphosphonium, antimicrobial

## Abstract

Biofouling is the buildup of marine organisms on a submerged material. This research tests the efficacy of phosphonium ion gels comprising phosphonium monomers ([P_444VB_][AOT] and [P_888VB_][AOT]) and free ionic liquid ([P_4448_][AOT], [P_88814_][AOT]) (10 to 50 wt%), varying copper(II) oxide biocide concentrations (0 to 2 wt%), and the docusate anion [AOT]^−^ for added hydrophobicity. The efficacy of these formulations was tested using a seachest simulator protected from light and tidal currents in New Zealand coastal waters over the summer and autumn periods. Anti-fouling performance was correlated with the hydrophobicity of the surface (water contact angle: 14–131°) and biocide concentration. Formulations with higher hydrophobicity (i.e., less free ionic liquid and longer alkyl chain substituents) displayed superior anti-fouling performance. The presence of the copper(II) biocide negatively affected anti-fouling performance via significant increases to hydrophilicity. No correlation was observed between antimicrobial activity and anti-fouling performance. Overall, phosphonium ion gels show potential for combining anti-fouling and foul release properties.

## 1. Introduction

Marine biofouling is defined as the undesirable accumulation of microbes, algae, and other marine animals on a submerged substrate [1]. When a non-toxic material is submerged in seawater, it will inevitably experience fouling to some extent. Thus, biofouling affects stationary underwater structures, ships, and other marine vessels [2]. Marine biofouling causes economic damage through the translocation of pests and invasive species, faster material degradation, reduced asset lifetime, and increases marine vessels’ fuel consumption by up to 40–50% [3,4,5].

The primary method to prevent (or reduce) biofouling is to apply anti-fouling coatings to the underwater surfaces of stationary objects and mobile vessels. Anti-fouling solutions are most commonly paints containing a chemical biocide, which can be applied to the outer surface of vessel hulls [1,6,7]. Anti-fouling coatings primarily act via the slow release of toxic components that prevent the initial adhesion and colonization stages of biofouling (Figure 1). Tributyl tin (TBT) [7], SLIPS coatings [8,9,10], biofilm-based ionic liquid coating [11], and hydrogels are some examples of current anti-fouling materials. These biocides also confer a ’halo’ defence, inhibiting biofouling in an area around the coating surface [9].

Seahawk Cukote [13], TotalBoat JD select (ablative) [13], Pettit Odyssey Triton (ablative) [13], Carboline Sea-barrier 1000 [14], and Carboline Sea-barrier 3000 [13] are examples of some commercially available anti-fouling paints. Although the most effective for preventing fouling, the biocides in these coatings have significant off-target environmental impacts. For example, the International Maritime Organization (IMO) banned TBT-based self-polishing co-polymers in 2008 because of the accumulation of organic tin compounds in fish, seabirds, and other marine species [7].

A second class of anti-fouling coatings comprises slippery liquid-infused porous surfaces (SLIPSs), which act by creating a slippery surface that limits the strength of biofouling adhesion. SLIPS materials include SLIPS^®^ Foul Protect^TM^ Marine Addictive, SLIPS^®^ Dolphin^TM^ Bottom Paint [10], and AkzoNobel Intersleek 1100SR [15]. SLIP materials are created by infusing microporous membranes with liquid lubricants [16,17]. Fouling species are detached when vessels are underway due to the high shear forces cleaving the foulants from the hull. While SLIPS coatings are viable for mobile vessels that spend a large proportion of time underway at sea, they are ineffective for recessed areas, such as sea chests, that do not experience high shear flows. Alongside the need for high shear flow to remove attached species, another limitation of a SLIPS is that it only prevents fouling on the surfaces where the SLIPS coating is applied. Unlike chemical biocides, SLIPSare not specifically designed to eliminate or repel attached or approaching microbes. As a result, fouling species can remain viable in the surrounding water and adhere to any surface not protected by SLIPS coating [16,18,19]. Consequently, any mechanical damage to the coating surface, e.g., during docking, anchoring, and encountering floating debris, will expose adhesion sites for fouling to occur [9].

Ionic liquids (ILs) and ion gels have yet to be widely explored for marine anti-fouling (Appendix A) [4,20,21]. For context, ionic liquids are salts that melt at or around room temperature (25 °C) [22,23]. Of interest to the anti-fouling application, ionic liquids can form ion gels similar to hydrogels but with surface chemistry and toxicity that can be tailored through the chemistry of the ionic liquid. There already exists significant academic and industrial interest in developing ionic liquids for preservative agents, disinfectants, surface protectants, and anti-fouling agents [6,24,25,26,27,28]. Ionic liquids with cationic components of imidazolium, pyridinium, ammonia, and phosphonium have been found to have strong antibacterial activity [24,29,30,31]. By choosing an antimicrobial anion, one can adjust an ionic liquid’s antimicrobial properties [32]. In a 2020 investigation, Wylie et al. looked into the antibacterial properties of SLIPSs injected with various counterions of phosphonium ionic liquids using different anions such as bis(trifluoromethyl sulfonyl)imide and docusate [9].

We perceive several advantages in using phosphonium-based ionic liquids for producing anti-fouling coatings. There are four possible alkyl chain attachments onto the phosphonium ion, [PR_4_]^+^, which can be used to significantly enhance hydrophobicity or toxicity depending on the appropriate choice of alkyl substituent. Additionally, phosphonium ionic liquids are thermally more stable than imidazolium and other common ionic liquids, more stable to bases and nucleophiles (no acidic protons), and less expensive [22,33]. Explored chemistries can provide phosphonium ionic liquids with improved heat stability, lower viscosity, improved gas transport, and greater stability under basic and reducing conditions [29,34].

In this work, we describe the synthesis and explore the use of hydrophobic phosphonium-based poly(ionic liquid) ion gels to prevent biofouling in marine environments. We observe the effect of hydrophobicity, ionic liquid content, and the inclusion of copper as a toxic component. We determine correlations between hydrophobicity and antimicrobial properties of the phosphonium ion gels with anti-fouling performance in marine environments.

## 2. Materials and Methods

### 2.1. Instrumentation and Measurement

NMR spectra were recorded on a JEOL 400 MHz NMR Spectrometer and Magritek Spinsolve 60 Ultra NMR Spectrometer. FT-IR spectra were recorded on a BRUKER Alpha II FT-IR Spectrometer. UV-Vis readings for micro testing were recorded on Biotek Cytation 5 Imaging Reader. Mass Spectra were recorded on Waters SYNAPT XS High-Resolution Mass Spectrometer. KSV CAM200 instrument was used for recording the Water Contact Angle. Surface roughness was recorded using Filmetrics Profilm3D and JEOL JSM IT-300 variable pressure scanning electron microscope.

### 2.2. General Procedure

All chemicals and solvents, unless otherwise noted, were bought from Sigma-Aldrich (St. Louis, MO, USA) and utilized as received.

#### 2.2.1. Tributyl(vinylbenzyl)phosphonium Chloride ([P_444VB_][Cl]) [35]

Vinylbenzyl chloride (0.075 mol, 11.45 g) was added dropwise to a vigorously stirred solution of tributylphosphine (0.061 mol, 12.34 g) in toluene (28 mL) at room temperature under a nitrogen atmosphere. The reaction mixture was stirred at room temperature for 40 h. The precipitated salt was collected, washed three times with toluene, and dried under vacuum (200 m Torr) (17.20 g, 79.45%). **^1^H NMR** (400 MHz, CDCl_3_) δ 7.48–6.96 (m, 5H, H_14_, H_15_, H_16_, H_17_), 6.54 (dd, J = 17.6, 10.8 Hz, 1H, H_18_), 5.64 (t, J = 17.5 Hz, 1H, H_20_), 5.15 (dd, J = 11.0, 2.5 Hz, 1H, H_19_), 2.77–1.83 (m, 10H, H_1_, H_5_, H_9_, H_13_), 1.33 (dq, J = 11.3, 6.5, 6.0 Hz, 13H, H_2_, H_3_, H_6_, H_7_, H_10_, H_11_), 1.03–0.54 (m, 9H, H_4_, H_8_, H_12_). **MS** (+ESI) (Acetonitrile): m/z 319.2550 [C_21_H_36_P] calc. 319.2555. (−ESI) (Acetonitrile): m/z Not Observed [Cl] calc. 34.9689.

#### 2.2.2. Tributyl(octyl)phosphonium Bromide ([P_4448_][Br]) [35,36]

1-bromooctane (0.061 mol, 13.17 g) was added dropwise to a vigorously stirred solution of tributylphosphine (0.061 mol, 12.50 g) in toluene (28 mL) at 100 ^°^C under a nitrogen atmosphere. The reaction mixture was stirred for 40 h. The sample was then evaporated under reduced pressure to remove excess toluene from the compound, and dried under vacuum (200 m Torr)(20.3129 g, 84.21%). **^1^H NMR** (400 MHz, CDCl_3_) δ 2.42–2.31 (m, 3H, H_1_, H_5_, H_9_), 2.35–2.27 (m, 1H, H_13_), 1.93 (s, 1H, H_14_), 1.47 (s, 2H, H_6_, H_7_), 1.53–1.44 (m, 2H, H_2_, H_3_), 1.44 (d, J = 7.6 Hz, 2H, H_10_, H_11_), 1.44–1.31 (m, 1H,), 1.29–1.13 (m, 2H, H_15_, H_16_), 1.17 (s, 3H, H_17_, H_18_, H_19_), 0.92–0.81 (m, 5H, H_4_,H_8_), 0.85–0.74 (m, 2H, H_12,_ H_20_). **MS** (+ESI) (Acetonitrile): m/z 315.3177 [C_20_H_44_P] calc. 315.3181. (−ESI) (Acetonitrile): m/z Not Observed [Br] calc. 78.9183.

#### 2.2.3. Tributyl(vinylbenzyl)phosphonium Docusate ([P_444VB_][AOT])

Sodium docusate ([Na][AOT]) (0.015 mol, 6.70 g) was dissolved in 30 mL of dichloromethane (DCM). The [Na][AoT] solution was added dropwise to a separate round-bottom flask containing [P_444VB_][Cl] (0.015 mol, 5.35 g) dissolved in 15 mL of DCM. The reaction mixture was stirred for 12 h at room temperature. The resulting suspension was then filtered through Celite. The filtrate was evaporated under reduced pressure and dried under high vacuum (200 m Torr) for 2 h to produce [P_444VB_][AOT] as a viscous pale-yellow liquid (8.78 g, 78.99%). **^1^H NMR** (400 MHz, CDCl_3_) δ 7.43–7.25 (m, 4H, H_14_, H_15_, H_16_, H_17_), 6.67 (ddd, J = 17.6, 11.0, 2.1 Hz, 1H, H_18_), 5.84–5.70 (m, 1H, H_19_), 5.33–5.22 (m, 1H, H_20_), 4.18 (dd, J = 11.4, 3.6 Hz, 1H), 4.08–3.84 (m, 5H, H_21_, H_22_, H_31_), 3.27 (dd, J = 17.5, 11.4 Hz, 1H, H_30_), 3.14 (dd, J = 17.5, 3.7 Hz, 1H, H_30_), 2.29–2.13 (m, 4H, H_1_, H_5_), 1.99 (s, 1H, H_23_), 1.60 (q, J = 5.7 Hz, 1H, H_32_), 1.56–1.19 (m, 18H, H_2_, H_3_, H_6_, H_7_, H_10_, H_11_, H_26_, H_27_, H_28_, H_24_, H_35_, H_36_, H_37_), 1.25 (s, 7H, H_24_, H_35_, H_36_, H_37_), 0.96–0.76 (m, 17H, H_4_, H_8_, H_12_, H_25_, H_29_, H_34_, H_38_). **FT-IR: v/cm^−1^**= 2931 (m,sh), 1630 (w,sh), 1512 (w,sh), 1731 (s,sh), 1323 (w,sh), 1158 (s,sh), 853 (s,sh), 990 (s,sh) 908 (s,sh). **MS** (+ESI) (Acetonitrile): m/z 319.2548 [C_21_H_36_P] calc. 319.2555. (−ESI) (Acetonitrile): m/z 421.2267 [C_20_H_37_O_7_S] calc. 421.2260.

#### 2.2.4. Tributyl(octyl)phosphonium Docusate ([P_4448_][AOT])

Sodium docusate (1.24 mmol, 0.55 g) was dissolved in 5 mL of DCM. The [Na][AoT] solution was added dropwise to a separate round-bottom flask containing [P_4448_][Br] (1.30 mmol, 0.51 g) dissolved in 10 mL of DCM. The reaction mixture was stirred for 12 h at room temperature. The resulting suspension was then filtered through Celite. The filtrate was evaporated under reduced pressure and dried under high vacuum (200 mTorr) for 2 h to produce [P_4448_][AOT] as a viscous cloudy-white liquid (0.70 g, 73.05%). **^1^H NMR** (400 MHz, CDCl_3_) δ 4.18–3.99 (m, 1H, H_22_), 4.02–3.86 (m, 1H, H_31_), 2.32 (s, 1H, H_13_), 2.38–2.23 (m, 1H, H_1_, H_5_), 2.11 (s, 1H, H_9_), 1.57–1.29 (m, 7H, H_26_, H_27_, H_28_), 1.32–1.18 (m, 8H, H_24_, H_35_, H_36_, H_37_), 1.02–0.89 (m, 4H, H_33_, H_2_), 0.93–0.85 (m, 1H, H_21_), 0.88–0.79 (m, 5H, H_4_, H_8_, H_12_, H_20_, H_29_, H_38_). **FT-IR: v/cm^−1^** = 2926 (s,sh), 1680 (s,sh), 1313 (w,sh), 1160 (w,sh), 810 (s,sh). **FT-IR: v/cm^−1^** = 2926 (s,sh), 1680 (s,sh), 1313 (w,sh), 1160 (w,sh), 810 (s,sh). **MS** (+ESI) (Acetonitrile): m/z 315.3175 [C_20_H_44_P] calc. 315.3181. (−ESI) (Acetonitrile): m/z 421.2268 [C_20_H_37_O_7_S] calc. 421.2260.

#### 2.2.5. Trioctyl(vinylbenzyl)phosphonium Chloride ([P_888VB_][Cl]) [35]

Vinylbenzyl chloride (0.068 mol, 10.38g) was added dropwise to a vigorously stirred solution of trioctylphosphine (0.068, 25.20 g) in acetonitrile (25 mL) at 60 °C under an atmosphere of nitrogen. The reaction mixture was stirred at room temperature for 24 h. The precipitated salt was collected, washed three times with acetonitrile, and dried under vacuum (200 m Torr) (33.55 g, 94.29%). **^1^H NMR** (62 MHz, CDCl3) δ 7.23 (q, J = 3.4, 2.4 Hz, 1H, H_26_, H_27_, H_28_, H_29_), 2.21 (dd, J = 15.0, 7.3 Hz, 1H, H_1_, H_9_, H_18_, H_25_), 1.23 (d, J = 5.2 Hz, 2H, H_2_, H_3_, H_4_, H_5_, H_6_, H_7_, H_10_, H_11_, H_12_, H_13_, H_14_, H_15_, H_18_, H_19_, H_20_, H_21_, H_22_, H_23_), 0.90 (d, J = 22.4 Hz, 5H, H_8_, H_16_, H_24_). **FT-IR: v/cm^−1^** = 2923 (s,sh), 1629 (w,sh), 1511 (w,sh), 989 (s,sh), 904 (s,sh). **FT-IR: v/cm^−1^** = 2923 (s,sh), 1630 (w,sh), 1511(w,sh), 989 (s,sh), 904 (s,sh). **MS** (+ESI) (Acetonitrile): m/z 487.4424 [C_33_H_60_P] calc. 487.4433. (−ESI) (Acetonitrile): m/z Not Observed [Cl] calc. 34.9689.

#### 2.2.6. Trioctyl(tetradecyl)phosphonium Bromide ([P_88814_][Br]) [35,36]

1-bromotetradecane (0.034 mol, 9.47 g) was added dropwise to a vigorously stirred solution of trioctylphosphine (0.034 mol, 12.46 g) in acetonitrile (25 mL) at 100 °C under a nitrogen atmosphere. The reaction mixture was stirred for 26 h. The sample was then evaporated under reduced pressure to remove excess acetonitrile from the compound and dried under a high vacuum (200 m Torr)(20.52 g, 93.26%). **^1^H NMR** (62 MHz, CDCl3) δ 2.78–1.94 (m, 0H, H_1_, H_9_, H_17_, H_25_), 1.56–0.51 (m, 4H, H_2_, H_3_, H_4_, H_5_, H_6_, H_7_, H_8_, H_10_, H_11_, H_12_, H_13_, H_14_, H_15_, H_16_, H_18_, H_19_, H_20_, H_21_, H_22_, H_23_, H_24_, H_26_, H_27_, H_28_, H_29_, H_30_, H_31_, H_32_, H_33_, H_34_, H_35_, H_36_, H_37_). **FT-IR: v/cm^−1^** = 2921 (s,sh). **FT-IR: v/cm^−1^** = 2921 (s,sh). **MS** (+ESI) (Acetonitrile): m/z 567.5990 [C_38_H_80_P] calc. 567.5998. (−ESI) (Acetonitrile): m/z Not Observed [Br] calc. 78.9183.

#### 2.2.7. Trioctyl(vinylbenzyl)phosphonium Docusate ([P_888VB_][AOT])

Sodium docusate (2.39 mmol, 1.06 g) was dissolved in 10 mL of DCM. The [Na][AOT] solution was added dropwise to a separate round-bottom flask containing [P_888VB_][Cl] (2.39 mmol, 1.25 g) dissolved in 10 mL of DCM. The reaction mixture was stirred for 12 h at room temperature. The resulting suspension was then filtered through Celite. The filtrate was evaporated under reduced pressure and dried under high vacuum for 2 h to produce [P_888VB_][AOT] as a viscous pale-yellow liquid (1.5 g, 73.62%). **^1^H NMR** (400 MHz, CDCl_3_) δ 7.43–7.25 (m, 4H, H_14_, H_15_, H_16_, H_17_), 6.67 (ddd, J = 17.6, 11.0, 2.1 Hz, 1H, H_18_), 5.84–5.70 (m, 1H, H_19_), 5.33–5.22 (m, 1H, H_20_), 4.18 (dd, J = 11.4, 3.6 Hz, 1H), 4.08–3.84 (m, 5H, H_21_, H_22_, H_31_), 3.27 (dd, J = 17.5, 11.4 Hz, 1H, H_30_), 3.14 (dd, J = 17.5, 3.7 Hz, 1H, H_30_), 2.29–2.13 (m, 4H, H_1_, H_5_), 1.99 (s, 1H, H_23_), 1.60 (q, J = 5.7 Hz, 1H, H_32_), 1.56–1.19 (m, 18H, H_2_, H_3_, H_6_, H_7_, H_10_, H_11_, H_26_, H_27_, H_28_, H_24_, H_35_, H_36_, H_37_), 1.25 (s, 7H, H_24_, H_35_, H_36_, H_37_), 0.96–0.76 (m, 17H, H_4_, H_8_, H_12_, H_25_, H_29_, H_34_, H_38_). **^13^C NMR** (101 MHz, CDCl_3_) δ 171.76, C_12_ 169.16, C_8_ 135.80 C_61_, C_64_, 130.36, 130.30, 129.63, 129.34, 129.29, 127.83, 127.17, 127.13, 126.09, 115.36, 114.88 C_65_, 67.57, 67.49, 66.89, 61.97, 53.40, 38.66, 38.64, 38.52, 38.46, 34.36, 31.66, 30.84, 30.70, 30.29, 30.25, 30.07, 30.01, 28.91, 28.87, 28.81, 27.01, 26.85, 26.56, 23.62, 23.38, 22.96, 22.93, 22.55, 21.80, 21.75, 18.85, 18.39, 14.05, 14.01, 10.92, 10.87, 10.76, 10.73. **FT-IR: v/cm^−1^**= 2926 (s,sh), 1630 (w,sh), 1512 (w,sh), 1732 (s,sh), 1325 (w,sh) 1155 (s,sh), 856 (s,sh), 989 (s,sh), 906 (s,sh). **MS** (+ESI) (Acetonitrile): m/z 487.4426 [C_33_H_60_P] calc. 487.4433. (−ESI) (Acetonitrile): m/z 421.2263 [C_20_H_37_O_7_S] calc. 421.2260

#### 2.2.8. Trioctyl(tetradecyl)phosphonium Docusate ([P_88814_][AOT])

Sodium docusate (4 mmol, 1.78 g) was dissolved in 10 mL of DCM. The [Na][AOT] solution was added dropwise to a separate round-bottom flask containing [P_88814_][Br] (4 mmol, 2.6 g) dissolved in 10 mL of DCM. The reaction mixture was stirred for 12 h at room temperature. The resulting suspension was then filtered through Celite. The filtrate was evaporated under reduced pressure and dried under high vacuum for 2 h to produce [P_88814_][AOT] as a viscous cloudy-white liquid (2.5961 g, 65.42%). **^1^H NMR** (400 MHz, CDCl_3_) δ 4.20–3.85 (m, 4H, H_40_, H_49_), 3.29–3.06 (m, 1H, H_39_), 2.36–2.22 (m, 7H, H_1_, H_9_, H_17_, H_25_), 2.02 (d, J = 27.5 Hz, 3H, H_41_, H_50_), 1.68–1.15 (m, 79H, H_2_, H_3_, H_4,_ H_5_, H_6_, H_7_, H_10_, H_11_, H_12_, H_13_, H_14_, H_15_, H_18_, H_19_, H_20_, H_21_, H_22_, H_23_, H_26_, H_27_, H_28_, H_29_, H_30_, H_31_, H_32_, H_33_, H_34_, H_35_, H_36_, H_37_, H_42,_ H_44_, H_45_, H_46_, H_51,_ H_53_, H_54_, H_55_), 0.93–0.79 (m, 24H, H_8_, H_16_, H_24_, H_38_, H_43_, H_47_, H_52_, H_56_). **^13^C NMR** (101 MHz, CDCl_3_) δ 171.72, C_51_, 169.27, C_47_, 67.02, C_41_, C_53_, 61.73, C_48_, 38.64, 38.52, 38.46, 34.05, 31.90, 31.69, 30.85, 30.70, 30.28, 30.25, 30.09, 30.04, 29.67, 29.63, 29.53, 29.34, 29.30, 29.02, 28.95, 28.87, 23.61, 23.40, 22.98, 22.95, 22.67, 22.58, 21.94, 21.89, 19.29, 18.83, 14.09, 14.07, 14.03, 10.91, 10.87, 10.82, 10.77. **FT-IR: v/cm^−1^** = 2923 (s,sh), 1733 (s,sh), 1352 (w,sh), 1157 (s,sh), 771 (s,sh). **MS** (+ESI) (Acetonitrile): m/z 567.5989 [C_38_H_80_P] calc. 567.5998. (−ESI) (Acetonitrile): m/z 421.2265 [C_20_H_37_O_7_S] calc. 421.2260.

### 2.3. Polymerization

To prepare ion gels, appropriate ratios of the polymerizing ionic liquids [P_xxxVB_][AOT] and free ionic liquid [P_xxxy_][AOT] were mixed with the crosslinker divinyl benzene (DVB), the photoinitiator 2-hydroxy-2-methylpropiophenone, and (if applicable) the biocide copper(II) oxide (CuO). The specific formulations prepared are summarized in Table 1 and Table 2 below. Each formulation was vortexed for 5 min, allowed to settle to remove bubbles, then coated over the stainless coupon (previously coated with commercial epoxy) using a roll coater. The coupon was then placed in a gas-tight curing chamber and flushed with nitrogen for 15 min before curing under UV-Vis lamb of irradiance 2.213–2.113 mW/cm^2^. Cured films were slightly sticky to the touch and ranged between 30 and 50 microns in thickness. The resulting coatings are referred to by their wt% content of polymerizable IL, free IL, and biocide, e.g., Pxxx[88,10,0], is trialkylphosphonium with 88 wt% polymerizable IL, 10 wt% free IL, and 0 wt% biocides, and the balance is crosslinker and photoinitiator.

### 2.4. Contact Angle

A film (approximately 150 mg; 20–30 microns in thickness) of each composition was cast on a glass plate. Advancing contact angle measurements were recorded using a CAM 200 apparatus and processed using CAM 2008 software. Advancing contact angles were measured, using 4 µL droplets of deionized water as a wetting agent, for 30 frames with an interval of one second for each formulation. Measurements were conducted at room temperature.

### 2.5. Field Testing

Samples were coated on an epoxy-treated stainless coupon provided by Defence Technology Agency (New Zealand) and sent for field testing at Devonport Naval Base. The samples were submerged in coastal seawater inside a seachest simulator that allowed the slow exchange of seawater inside a dark environment protected from strong currents and flow. Control samples were included inside the seachest simulator and inspected at regular intervals for biofouling accumulation.

### 2.6. Microbiology Testing

Escherichia coli (K12) (Gram-negative) and Staphylococcus Epidermidis (Gram-positive) were cultured and received from the School of Biological Science, University of Canterbury, New Zealand.

Antibacterial activity was investigated by examining Escherichia coli (K12) and Staphylococcus epidermidis biofilm formation over the PILs. A single bacteria colony was incubated overnight in Luria Bertani (LB) broth. The optical density at 600 nm (OD600) of each bacterium was determined and a solution of 0.1 OD was prepared in LB. This was the stock solution for the testing.

#### 2.6.1. Fluorescence Assay

Following established methods [37,38], phosphonium ion gel samples were coated on a Whatman No 1 Filter paper, then cut into 5 mm discs, transferred to a 96-well plate containing 100 μL of the bacteria solutions described above, and incubated at 37 °C. After 24 h, each disc was gently washed with phosphate-buffered saline (PBS) solution and incubated with 10% Alamar Blue at 37 °C for 2 h. The microorganisms’ viability was then measured via fluorescence in a spectrophotometer with excitation at 540 nm and emission at 590 nm. Biofilm inhibition was then calculated considering the reference disc of filter paper (denoted Control) as an example of 100% viability (zero inhibition). All experiments were performed in five replicates.

#### 2.6.2. Disc Diffusion Assay

Following established methods [9,39], phosphonium ion gel samples were coated on a Whatman No 1 Filter paper and cut into 5 mm discs, transferred to an agar plate containing 100 μL of the bacteria solution described above spread evenly across the plate, and incubated at 37 °C for 24 h to allow the bacteria to grow and the antibacterial agent to diffuse into the agar. After incubation, the diameter of the zone of inhibition was measured. The size of the zone was proportional to the effectiveness of the antibacterial agent against the bacterial strain. Biofilm inhibition was then calculated, considering the reference disc (denoted Control) as having 100% viability (zero inhibition). All the experiments were performed in triplicate.

## 3. Result and Discussion

### 3.1. Preparation of Phosphonium Ion Gel Coatings

#### 3.1.1. Preparation of Phosphonium Docusate Ionic Liquids

The class of phosphonium docusate ionic liquids was chosen for study because it allowed a combination of the phosphonium cation, known to have antimicrobial activity and four alkyl substituents that can convey significant hydrophobicity [29,40,41] with the docusate anion, which has been shown to be hydrophobic and possess antimicrobial properties [9,34]. This led to the hypothesis that the combination would produce ionic liquids and ion gels with marine anti-fouling properties similar to both biocide-containing polymers and hydrophobic SLIPS materials.

The docusate ionic liquids were prepared via the metathesis of the known halide intermediates tributyl(vinylbenzyl)phosphonium chloride [35], trioctyl(vinylbenzyl)phosphonium chloride [35], tributyl(octyl)phosphonium bromide [35,36], and trioctyl(tetradecyl)phosphonium bromide (Figure 2) [35,36]. The metathesis reaction was adapted from that described by Wylie et al., 2020 for the metathesis of phosphonium ionic liquids [9].

The monomers and free ionic liquids were characterized through ^1^H and ^13^C NMR (cation structure), FT-IR (presence of docusate through C=O stretch and vinyl groups through C=C stretch), and Mass Spectrometry (confirming presence of cation and anion structures of appropriate molecular weights) (Appendix A). The removal of halide was confirmed using a silver nitrate test.

#### 3.1.2. Casting of Phosphonium Ion Gel Coatings

Mixtures of monomer and free phosphonium docusate ionic liquids combined with crosslinking agent divinylbenzene, photoinitiator (2-hydroxy-2-methylpropiophenone), and biocide copper(II) oxide (if required) were able to be cast without a solvent. The resulting solutions were easily coated uniformly onto stainless steel coupons that had been pre-treated with a commercial epoxy coating for corrosion inhibition.

The resulting gels were crosslinked through radical polymerization under an inert environment following coating. This method has been used to produce similar phosphonium ion gels for gas separations and ionic conductivity [42]. While it is difficult to scale it industrially, this approach to producing robust crosslinked films was judged sufficient for this proof-of-concept work. The polymerization was confirmed through FT-IR spectra, which showed that the C=C stretch between 1640–1610 cm^−1^ was missing after curing (Appendix A).

The exact compositions of the coatings are summarized in Table 1 and Table 2. The rationale behind selecting a wide variety of compositions was to test the marine anti-fouling effects of hydrophobicity via alkyl substituent length on the phosphonium cation (either C4 or C8), the liquid or gel-like morphology via the loading of free ionic liquid, and the biocide concentration via copper(II) oxide).

### 3.2. Field Testing Results

Field testing results are illustrated and summarized in Figure 3, Figure 4, Figure 5, Figure 6, Figure 7, Figure 8, Figure 9 and Figure 10. These results were obtained by arranging the coated coupons inside a seachest simulator to allow the slow exchange of seawater while providing darkness and protecting the coupons from tidal forces and currents. This methodology was chosen to maximize the fouling potential and simulate worst-case fouling conditions. The fouling experiments were performed over the summer and autumn (December to June 2023) with the seachest simulator submersed in coastal waters at a pier off the Devonport Naval Base in Auckland, New Zealand. Control samples were also included, and biofouling was monitored by briefly removing the seachest and documenting the biofouling buildup at regular (1-to-2-week) intervals.

#### 3.2.1. Best-Performing Anti-Fouling Phosphonium Ion Gels

Figure 3 directly compares the anti-fouling performance of the best trioctyl and tributyl phosphonium ion gels. Visual observation shows that the P_888_ formulations outperform both the P_888_ with biocide and the P_444_ formulations.

Anti-fouling performance was quantified using the fouling rate (FR) standard, as guided by the Naval Ships’ Technical Manual Chapter 081 (Figure 4) [43]. The leading ion gel material produced in this work, P_888_[84,10,0], displays only 40% fouling coverage after 43 days submersed, with a maximum FR of 30. By day 63, the surface was 100% covered, but only led to an FR of 30. In contrast, the epoxy controls showed 80% coverage after 43 days and 100% coverage after 63 days with established tunicate, increasing the FR to 100. In contrast, P_888_[65,28,0] and other trioctylphosphonium samples that included biocide exhibited 100% coverage at FR 40 after 92 days. However, trioctylphosphonium samples without biocide (e.g., P_888_[93,0,0], P_888_[84,10,0], and P_888_[48,48,0]) showed a higher fouling rate, FR 100, at day 92.

Although these performances are inferior to commercial anti-fouling coatings (which contain copper biocides) [10], this work aims to understand the factors that govern the anti-fouling performance of phosphonium ion gel materials. Discovering that the presence of biocide resulted in higher fouling coverage and FR after equivalent time prompted further investigation.

#### 3.2.2. Effect of Hydrophobicity and Toxicity (via Alkyl Chain Length and Biocide Inclusion) on Anti-Fouling Performance

The fouling coverage and FR performance of all ion gels tested in this study are summarized in Figure 4, Figure 7 and Figure 10 and shown in Figure 5, Figure 8 and Figure 9.

The superior performance of the P_888_ ion gels (Figure 8) vs. the P_444_ ion gels (Figure 5) is clearly shown by comparing the FR graphs (Figure 4 and Figure 7). The P_444_ formulations received rapid colonization by fouling species, similar to the control sample of bare epoxy coating. Although the P_444_ showed a somewhat reduced fouling after 56 days, all P_444_ formulations were contaminated with excessive tube worm growth.

In contrast, no P_888_ formulations were contaminated with matured tube worms even after 63 days (Figure 6), although the FR had reached 100% surface coverage and parts of the coupons were judged as having an FR of 30, which is when cleaning for biosecurity reasons is required.

The superior performance of the more hydrophobic P_888_ materials was expected from the 2003 study conducted by Pernak et al. In that work, the authors examined the antibacterial and antifungal effects of imidazolium ionic liquids with alkyl substituent lengths ranging from 6 to 16 [23]. Their results showed that increasing alkoxy chain lengths was more effective for antimicrobial inhibition, up to a limit of 12–14 carbon atoms, which was the most effective carbon chain length against bacteria, fungi, and rods. Similarly, Metelytsia et al., 2022 studied the antibacterial properties of trialkylphosphonium ionic liquid by varying the alkyl chain length [32].

In contrast, it was a surprise to observe that coatings containing the copper(II) oxide biocide (Figure 10) had worse anti-fouling performance than the equivalent plain ion gels (Figure 4). Copper(I) and copper(II) oxides are often added to anti-fouling coating to inhibit the microbial growth that makes up the first stages of conditioning film and bacterial biofilm that initiate the fouling process (Figure 1).

In this work, coatings containing P_888_ ion gels without biocides (Figure 8) showed superior anti-fouling performance compared to P_888_ ion gels with biocides (Figure 9). P_888_ ion gels with biocides showed denser coverage at FR 10–30 on days 14, 43, and 63. We hypothesized that this was due to the copper(II) oxide causing a lower hydrophobicity in the coating. Advancing contact angle measurements were used to characterize the hydrophobicity of the surfaces. Tributylphosphonium- and trioctylphosphonium-sample advance contact angle measurements are summarised in Table 3 (Appendix A). It is evident from the table that the contact angle increases as the trialkylphosphonium’s attached alkyl chain length increases.

It is worth noticing that [AOT]^−^ falls within the category of anionic surfactants [9]. These compounds are best characterized by their negatively charged ions. Because of these ions, AOT can interact with water molecules and other polar substances, reducing surface tension. P_444_[77,21,0] shows the highest contact angle for tributyl phosphonium samples whereas P_444_[49,48,0] shows the least. This indicates that the contact angle degree decreases with an increased weight percentage of free ionic liquid in the formulation. This trend agrees with all the trioctyl samples, including the one having Cu^2+^ in it. The contact angle values [P_888VB_][AOT] were doubled as compared to [P_444VB_][AOT], proving that trioctyl(vinylbenzyl)phosphonium docusate is more hydrophobic than tributyl(vinylbenzyl)phosphonium docusate. Adding Cu^2+^ to [P_888VB_][AOT] reduces the contact angle to a more hydrophilic end.

#### 3.2.3. Correlation of Free Ionic Liquid Content and Marine Anti-Fouling Performance

The total fouling rate of each ion gel at the end of the trial period (53 and 63 days for the P_444VB_ and P_888VB_ ion gels, respectively) was calculated according to the procedure described in Appendix A and summarized in Figure 11. The clear difference between the control samples highlights the different rates of marine fouling under different conditions and the requirement for control samples.

The findings of the combined investigations show that the growth of biofilm formation is somewhat inhibited by both the tributyl and trioctyl formulations, with a reduction of about 50% relative to their controls (Figure 11). This implies that these formulations successfully reduce marine fouling.

The fouling rate for P_444_, P_888_, and P_888_ + biocide formulations was similar for all compositions of ionic liquid studied. While the proportion of ionic liquid is useful for controlling the durability and castability of such formulations, the increased mobility of the free ionic liquids, compared to the tethered ionic liquid moieties in the polymer backbone, does not play a role in anti-fouling performance for the early stages of fouling studied in this work.

In contrast, the hydrophobicity of the formulation (measured via contact angle) plays a clear role in anti-fouling performance (Figure 12). As shown, the trioctylphosphonium samples display contact angles from 83 to 131° and total fouling rates of 10–15% after 63 days whereas tributylphosphonium only achieves 13–60° and has higher total fouling rates (15–30%) after 63 days. It is promising that these relatively simple trialkylphosphonium materials can achieve such high hydrophobicity and anti-fouling performance through manipulation of the alkyl substituents. The hydrophobic action is similar to foul-release coatings such as SLIPS coatings [9].

#### 3.2.4. Correlation of Anti-Microbial Activity and Marine Anti-Fouling Performance

Antimicrobial testing was performed to determine whether antimicrobial resistance was linked to the prevention of marine anti-fouling performance. The widely accepted mechanism for marine fouling shown in Figure 1 considers bacterial adhesion the second stage of biofouling [1]. Therefore, we would expect that superior anti-fouling performance would be achieved with antibacterial materials. To study the inhibition capacity of the ionic liquids, trioctylphosphonium samples were exposed to both *S. epidermidis* (Gram-positive) and *E. coli* (Gram-negative), and observations of fluorescence and disc diffusion assays were recorded (Appendix A).

Figure 13 and Figure 14 shows the fluorescence readings of the trioctylphosphonium samples against *E. coli* and *S. epidermidis* after 24 h of incubation. All the samples showed inhibition against *S. epidermidis* (Gram-positive), with P_888_[65,28,0] and P_888_[48,48,0] showing the largest inhibition. In contrast, the phosphonium ion gels showed zero inhibition towards *E. coli.* As with the marine anti-fouling, the P_888_ series of ion gels showed the most inhibition of *S. epidermidis* (<50% the growth of the control sample). Interestingly, both ion gel formulations assisted the growth of the Gram-negative *E. coli,* which may have been due to providing phosphorous as a nutrient. These results suggest that antibacterial behaviour (towards Gram-positive bacteria) may contribute to anti-fouling performance. However, the results from disk diffusions assays were less clear. With the exception of sample P_888_[48,48,0], all formulations of the ion gels were ineffective against *E. coli* and *S. epidermidis* (Appendix A).

Disk diffusion assays were performed on all the coated samples using S. *epidermidis* and *E. coli*. The presence of a halo around the sample P_888_[48,48,0], which was dipped in a Gram-positive bacteria agar plate, concludes that the formulations are effective against Gram-positive bacteria. No halo was present around the samples placed in the Gram-negative bacteria plate.

The findings from this work somewhat reflect the 2020 study conducted by Wylie et al., who examined the antimicrobial behaviour of more aggressively hydrophobic trialkylphosphonium docusate ion gels [9]. The more hydrophobic ion gels were shown to be effective against both Gram-positive and Gram-negative bacteria (*S. aureus* and *P. aeruginosa*, respectively). The antibacterial behaviour of both phosphonium and imidazolium ionic liquids appears closely tied to the chain lengths of the alkyl substituents [23,44,45,46].

## 4. Conclusions

This work has provided proof-of-concept results for the application of phosphonium ion gel films to anti-fouling coatings for marine environments. New tributyl(vinylbenzyl)phosphonium docusate and trioctyl(vinylbenzyl)phosphonium docusate monomers were synthesized and used to produce ion gel coatings on epoxy-treated stainless-steel coupons. Ion gels of polymerized trioctyl(vinylbenzyl)phosphonium docusate performed best, with standardized foul ratings only reaching FR30 after 63 days. To emphasize, this foul rating performance was obtained in coastal waters and protected from light and water currents using anti-fouling coatings that did not contain biocide components.

The combination of strongly hydrophobic cation and anion components provides a possible route to metal-free anti-fouling coatings that would reduce the heavy metal contamination that currently results from anti-fouling coatings containing copper compounds.

The leading performance of the trioctyl(vinylbenzyl)phosphonium docusate ion gels was attributed to the increased hydrophobicity of the surfaces, with advancing contact angle measurements showing water contact angles >(83–131)°, up to double those of ion gels containing copper(II) oxide (<14–87°) and tributyl(vinyl benzyl)phosphonium docusate (<25–57°). It was found that the addition of copper(II) oxide as a biocide and free ionic liquid contributed to much lower hydrophobicities and correlated to significantly worse anti-fouling performance.

Interestingly, fluorescence assays indicated that marine anti-fouling performance correlated with the inhibition of Gram-positive bacteria. Further experimental work should aim to determine the link between the inhibition of the initial bacterial biofilm and long-term marine anti-fouling performance.

This work illustrates the promise of phosphonium-containing ion gel materials and provides clear design strategies for the development of phosphonium-containing ion gel materials as anti-fouling coatings for marine environments.

## Figures and Tables

**Figure 1 polymers-15-03677-f001:**
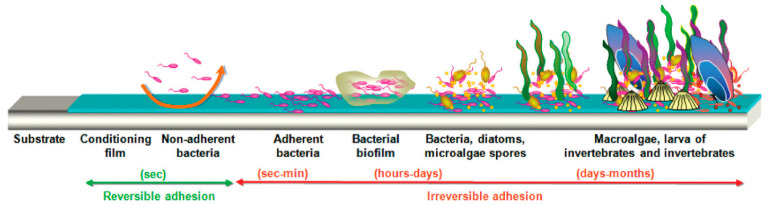
Process of marine biological fouling—Stage 1: Biofilm formation (by adsorbing organic molecules to the submerged surface, a conditioning biofilm is created); Stage 2: Primary colonization (a planktonic bacteria settles on the surface and grows exponentially in the nutrient-rich media); Stage 3: Secondary Colonization (colonization of microorganism including algal spores, barnacle cyprids and diatoms); Stage 4: Macro fouling formation (settlement of larger marine species) [1]. Figure has been reproduced with permission from reference [12].

**Figure 2 polymers-15-03677-f002:**
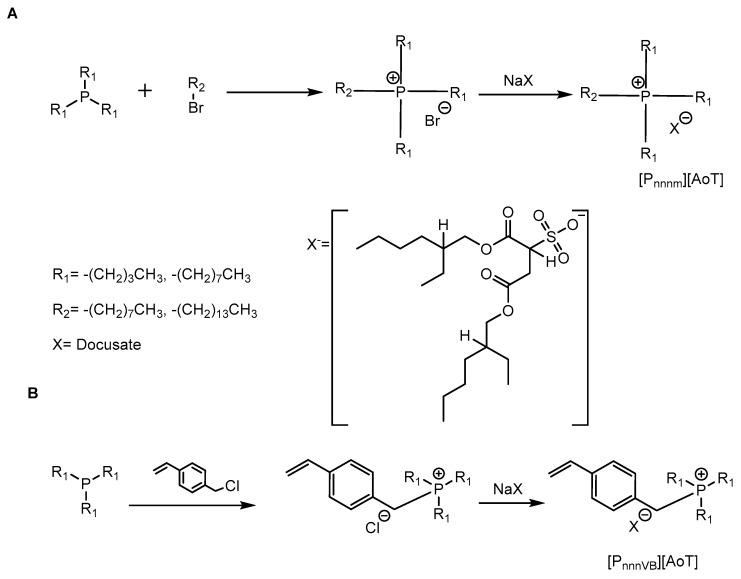
Synthesis schemes for preparing (**A**) polymerizing phosphonium ILs and (**B**) the phosphonium ILs.

**Figure 3 polymers-15-03677-f003:**
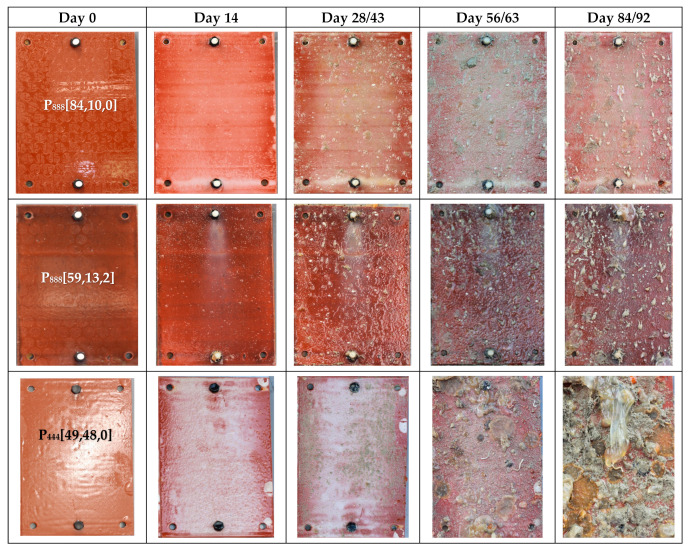
Comparison results of tributyl, trioctyl, and trioctyl with copper formulation after field testing in the seachest on Day 0, Day 14, Day 28/43, Day 56/63, and Day 84/92.

**Figure 4 polymers-15-03677-f004:**
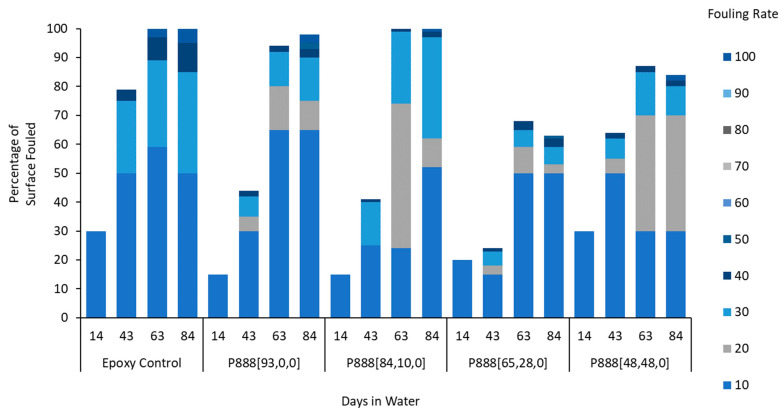
Field test results of trioctyl(vinylbenzyl)phosphonium docusate in water. The samples, labeled P888[a,b,c], were graphed in the order of increasing amounts of free ionic liquid in the formulation.

**Figure 5 polymers-15-03677-f005:**
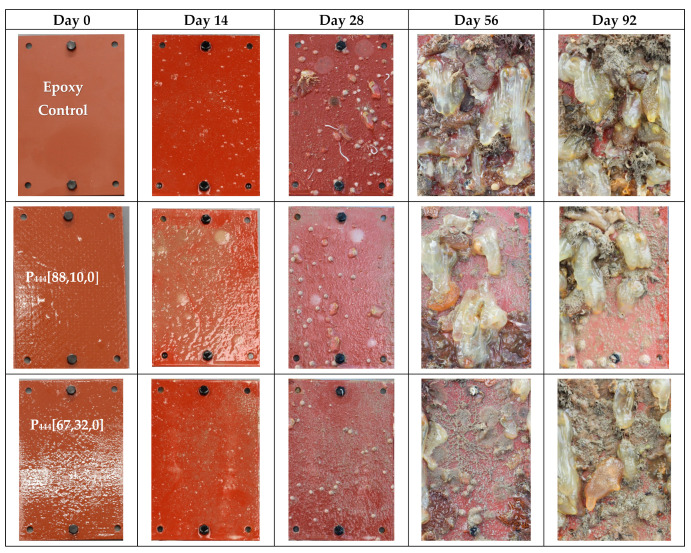
Field trial pictures of selected tributyl(vinylbenzy)phosphonium docusate formulation along with an epoxy-treated steel coupon as control on Day 0, Day 14, Day 28, Day 56, and Day 92. These coupons were tested in New Zealand’s coastal waters from December 2022 to February 2023. (Full data can be found in Appendix A).

**Figure 6 polymers-15-03677-f006:**
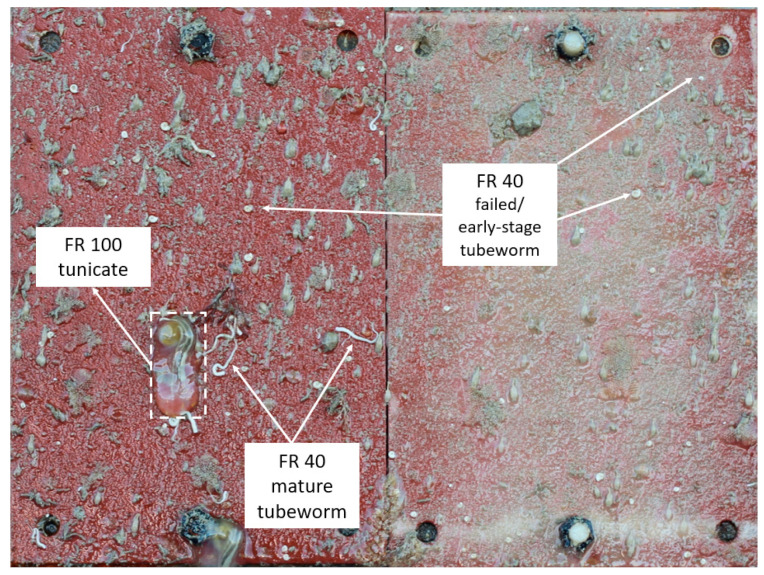
Comparison images of P_888_[84,10,0] (**right**) with the epoxy control (**left**) on Day 63. Although P_888_[84,10,0] is being colonized by tube worms (FR40), it inhibits the growth of tunicate (FR100), which is growing on the epoxy control.

**Figure 7 polymers-15-03677-f007:**
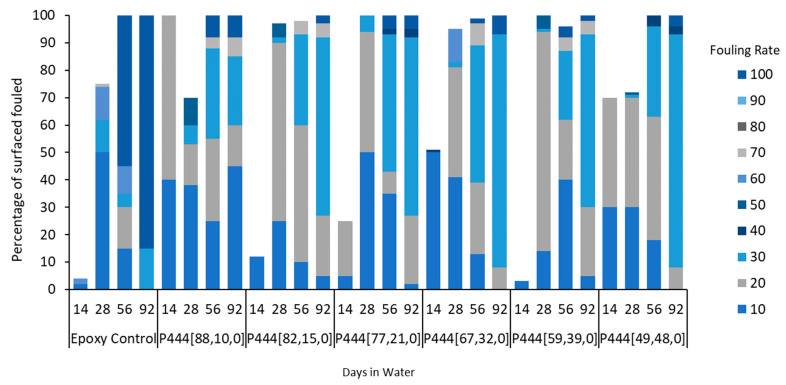
Field test results of tributyl(vinylbenzyl)phosphonium docusate in coastal seawater. The samples, labeled P_444_[a,b,c], are graphed in the order of increasing amounts of free ionic liquid in the formulation.

**Figure 8 polymers-15-03677-f008:**
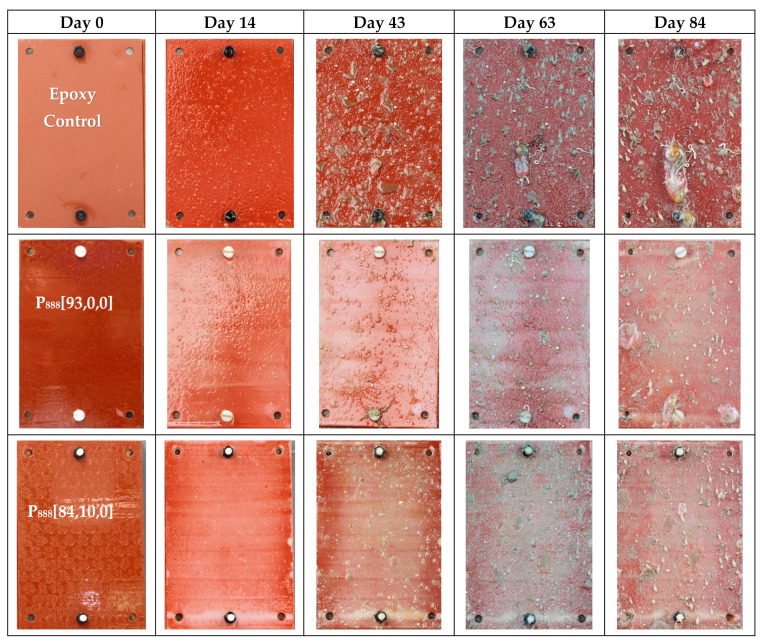
Field trial pictures of trioctyl(vinylbenzyl)phosphonium docusate formulation along with an epoxy-treated stainless steel coupon as control on Day 0, Day 14, Day 43, Day 63, and Day 84. These coupons were tested in New Zealand’s coastal waters from April 2023 to June 2023.

**Figure 9 polymers-15-03677-f009:**
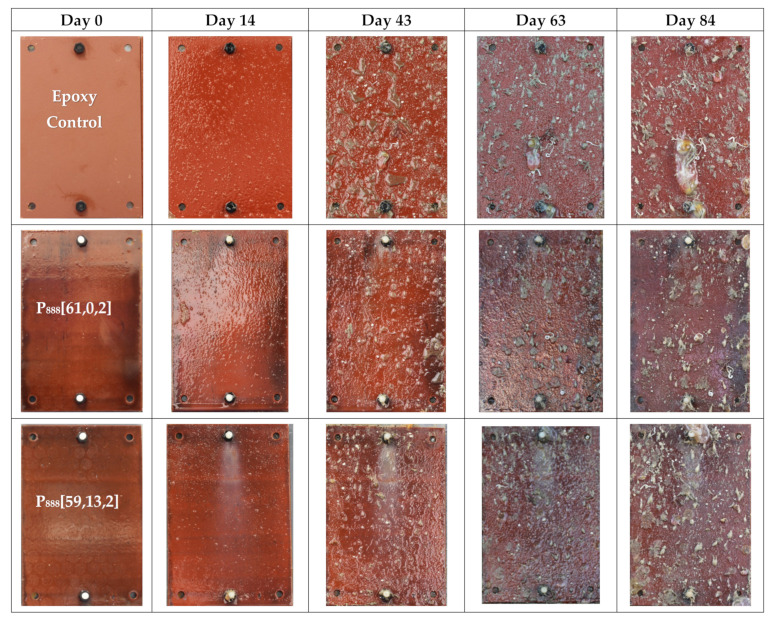
Field trial pictures of trioctyl(vinylbenzyl)phosphonium docusate formulation with Cu(II)O (biocide) along with an epoxy-treated stainless steel coupon as control on Day 0, Day 14, Day 43, Day 63, and Day 84. These coupons were tested in New Zealand’s coastal waters from April 2023 to June 2023.

**Figure 10 polymers-15-03677-f010:**
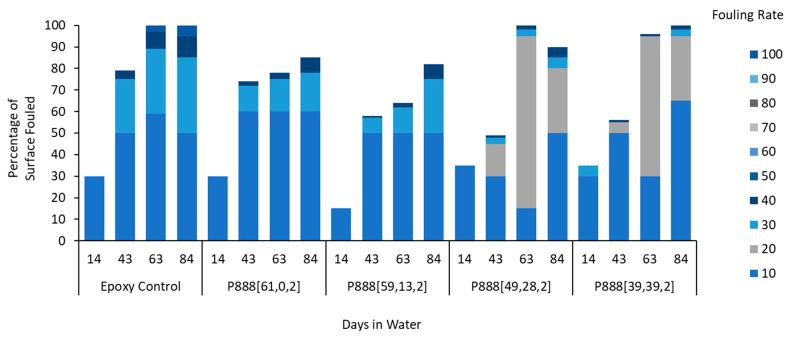
Field test results of trioctyl(vinylbenzyl)phosphonium docusate with copper oxide in water. The samples, labeled P888[a,b,c], were graphed in the order of increasing amounts of free ionic liquid in the formulation.

**Figure 11 polymers-15-03677-f011:**
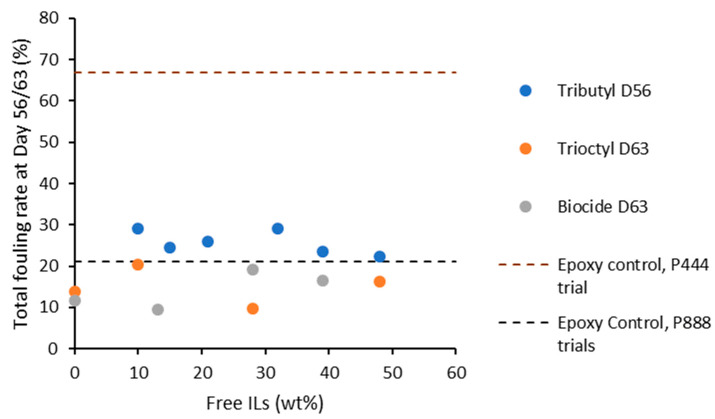
Total fouling rates of tributyl(vinylbenzyl)phosphonium docusate on Day 56 and trioctyl(vinylbenzyl)phosphonium docusate and trioctyl(vinylbenzyl)phosphonium docusate with biocide on Day 63 with respect to the control samples.

**Figure 12 polymers-15-03677-f012:**
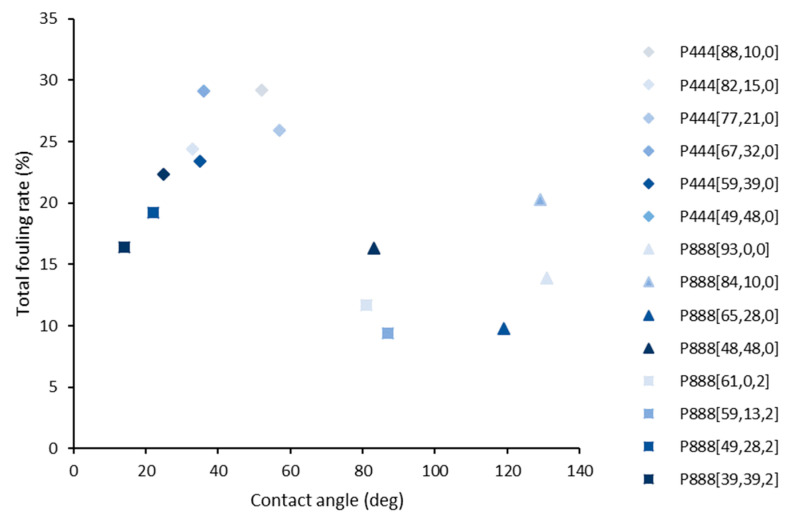
Graph showing relationship between water contact angle and total fouling rate. The samples are plotted in the decreasing order of free ionic liquid in the formulation. The color gradient (higher free ionic liquid content) is used to differentiate different formulations in trialkylphosphonium ionic liquid gels.

**Figure 13 polymers-15-03677-f013:**
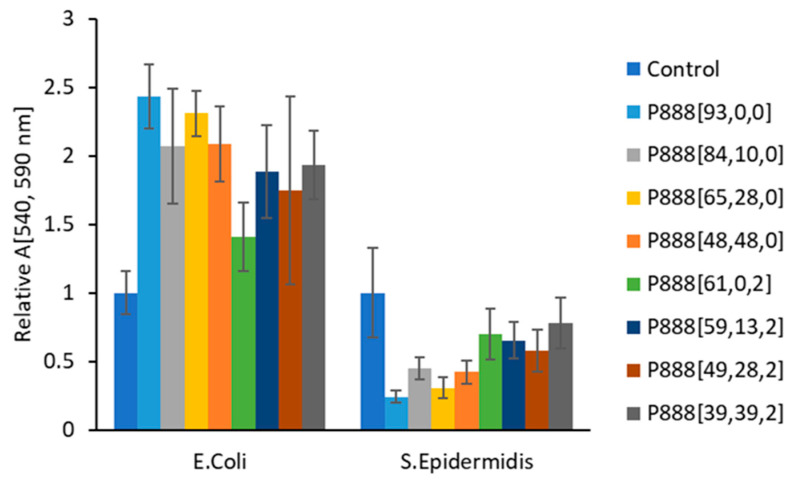
The graph shows the relative fluorescence readings of trioctylphosponium doocusate samples with respect to the control sample (where no bacterial growth was inhibited), recorded by exciting the samples at 540 nm and measuring the emissionat 585 nm. The sample labeled P_888_[a,b,c], where ‘a’ represents the percentage of the polymerizing group, ‘b’ represents the percentage of free ionic liquid, and ‘c’ represents the percentage of copper oxide used in the formulation, was subjected to treatments with both Gram-negative and Gram-positive bacteria. The error bars represent the relative standard deviation of the samples. The samples inhibited the growth of S. epidermidis, as the relative absorbance values in the figure were smaller than that of the control sample used, whereas there was zero inhibition of *E. coli* by the samples.

**Figure 14 polymers-15-03677-f014:**
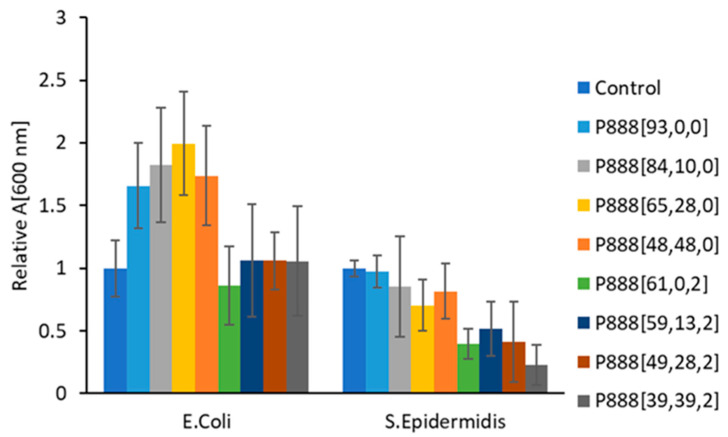
The graph shows the relative absorbance readings at 600 nm of trioctylphosponium doocusate samples with respect to the control sample (where no bacterial growth was inhibited). The sample labeled P_888_[a,b,c], where ‘a’ represents the percentage of the polymerizing group, ‘b’ represents the percentage of free ionic liquid, and ‘c’ represents the percentage of copper oxide used in the formulation, was subjected to treatments with both Gram-negative and Gram-positive bacteria. The error bars represent the relative standard deviation of the samples with respect to the control sample. The samples inhibited the growth of S. epidermidis, as the relative absorbance values were smaller than that of the control sample used, whereas there was zero inhibition of *E. coli* by the samples.

**Table 1 polymers-15-03677-t001:** The formulation for tributylphosphine used is denoted as P_444_[a,b,c], where ‘a’ represents the percentage of the polymerizing group, ‘b’ represents the percentage of free ionic liquid, and ‘c’ represents the percentage of copper oxide used in the formulation.

Formulation	[P_444VB_][AOT] (WT%)	[P_4448_][AOT] (WT%)	DVB (WT%)	Photoinitiator (WT%)	CuO (WT%)
P_444_[88,10,0]	88.10	10.03	1.27	0.60	0
P_444_[82,15,0]	81.96	14.50	2.35	1.19	0
P_444_[77,21,0]	76.78	20.86	1.18	1.18	0
P_444_[67,32,0]	66.50	31.5	1.25	0.75	0
P_444_[59,39,0]	59.19	38.78	1.03	1.00	0
P_444_[49,48,0]	49.00	48.00	2.00	1.00	0

**Table 2 polymers-15-03677-t002:** The formulation of trioctylphosphine is indicated by the label P_888_[a,b,c], where ‘a’ represents the percentage of the polymerizing group, ‘b’ represents the percentage of free ionic liquid, and ‘c’ represents the percentage of copper oxide used in the formulation.

Formulation	[P_888VB_][AOT] (WT%)	[P_88814_][AOT] (WT%)	DVB (WT%)	Photoinitiator (WT%)	CuO (WT%)
P_888_[93,0,0]	93.24	0	4.40	2.36	0
P_888_[84,10,0]	84.02	9.56	2.93	3.49	0
P_888_[65,28,0]	65.40	28.50	3.31	2.79	0
P_888_[48,48,0]	47.72	47.73	2.06	2.52	0
P_888_[61,0,2]	61.10	0	24.87	12.03	2.00
P_888_[59,13,2]	58.69	13.06	13.19	12.97	2.09
P_888_[49,28,2]	48.99	28.48	9.77	10.76	2.00
P_888_[39,39,2]	38.66	38.92	11.76	8.31	2.35

**Table 3 polymers-15-03677-t003:** Advancing water contact angle measurement of tributylphosphonium (left) and trioctylphosphonium (right) ion gels. The formulations of tributylphosphonium and trioctylphosphonium are indicated by the labels P_444_[a,b,c] and P_888_[a,b,c], where ‘a’ represents the percentage of the polymerizing group, ‘b’ represents the percentage of free ionic liquid, and ‘c’ represents the percentage of copper oxide used in the formulation.

Sample	Contact Angle (DEG)	Sample	Contact Angle (DEG)
P_444_[88,10,0]	52 ± 3	P_888_[93,0,0]	131 ± 8
P_444_[82,15,0]	33 ± 2	P_888_[84,10,0]	129 ± 8
P_444_[77,21,0]	57 ± 3	P_888_[65,28,0]	119 ± 4
P_444_[67,32,0]	36 ± 5	P_888_[48,48,0]	83 ± 4
P_444_[59,39,0]	35 ± 3	P_888_[61,0,2]	81 ± 1
P_444_[49,48,0]	25 ± 2	P_888_[59,13,2]	87 ± 2
		P_888_[49,28,2]	22 ± 4
		P_888_[39,39,2]	14 ± 3

## Data Availability

The data presented in this study are available as part of the Appendix A.

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
