# Peer review of "Anti-Fouling Properties of Phosphonium Ionic Liquid Coatings in the Marine Environment"

_polymers, 2023, doi:10.3390/polym15183677_

Round 1

Reviewer 1 Report

The authors report the efficacy of phosphonium ion gels comprised of phosphonium monomers ([P444VB][AOT] and [P888VB][AOT]) and free ionic liquid ([P4444][AOT], [P8888][AOT]) (10 to 50 wt%), varying lengths of alkyl chains (n = 4 and 8) and copper(II) oxide biocide concentration (0 to 2 wt%), and the docusate anion [AOT]- for added hydrophobicity. Formulations with higher hydrophobicity displayed superior antifouling performance. Presence of the Cu2+ biocide negatively affected antifouling performance via significant increases to hydrophilicity. The article is interesting. However, the characterization of gels is still enough. Some points of the manuscript also should be improved.

1.   The authors should carefully check the 1H NMR in the supporting information due to something wrong for the chemical shift of molecules. The authors are suggested to change the image of 1H NMR of molecules to clearly see the hydrogen atom assignment of the chemical shift.

2.   The gels should be further measured such as FTIR, XPS and SEM.

3.   The authors add the statistical bias in Figure 14 and other Figures.

4.   The formulation and sample in figures are inconsistent. Please check this problem.

5.   The anti-microbial activity of gel should be compared with other gels with similar components.

6.   Please carefully check the manuscript for writing and grammar.

Minor editing of English language required

Reviewer 2 Report

The introduction section should highlight the innovation and novelty of this study in greater detail.

What are the advantages of using tributyl(vinylbenzyl)phosphonium docusate and trioctyl(vinylbenzyl)phosphonium docusate compared to other antibacterial materials, in terms of bacterial killing efficacy, cost control, and potential for commercial scale-up?

Regarding the alkyl chain length, the authors only tested two variations, 4 and 8, which is insufficient to draw a solid conclusion about the relationship between antibacterial efficacy and carbon chain length.

There are numerous factors that could lead to differences in the contact angle. Even with similar materials, the variation in surface texture and roughness can influence the apparent contact angle. To eliminate these possibilities and understand the cause of the differences in contact angle among each sample, additional surface characterizations and surface roughness measurements need to be included.

Reviewer 3 Report

In this paper, the authors have described the synthesis, characterization and possible utility of phosphonium ion gel films for use as anti-fouling coatings for the marine environment. They have presented a thorough evaluation of the antifouling performance of such formulations, with a strong focus on hydrophobicity and the factors that impact it. This kind of work is within the scope of the Polymers journal. I admire the efforts of the authors in creating logical and extensive experiments in their work, including contact angle and microbiological analyses. It was interesting to see that the presence of copper biocide yielded surprising fouling results. Despite the fact that the antifouling performance was not markedly improved compared to existing solutions, this experimental work provides a useful insight into the factors affecting fouling and the strategies that others may utilize to improve upon this work. I recommend the paper for publication with a few revisions:

· There are some results observed have not been explained, like the disk diffusion assay outcomes, or the reason for the higher standard deviation with the observed contact angles for coatings with higher percentage of polymerizing group in trioctylphosphonium ion gels.

·  Statistical analysis would be helpful in determining if the data is indeed significantly different with differing a, b and c for the phosphonium ion gels

· The authors attributed the presence of copper biocide to lower hydrophobicity which makes sense, but it is unclear why it would perform so poorly against fouling, since copper biocides are commonly utilized as a toxic component to fouling. The formation of cuprous oxide upon immersion in the seawater should inhibit the settlement of fouling organisms. This could be explained further – perhaps it is due to conditions inside the seachest simulator?

· Reference 42 needs to be corrected
